# Network change point localisation under local differential privacy

**Mengchu Li**
Department of Statistics
University of Warwick
mengchu.li@warwick.ac.uk

**Thomas B. Berrett**
Department of Statistics
University of Warwick
tom.berrett@warwick.ac.uk

**Yi Yu**
Department of Statistics
University of Warwick
yi.yu.2@warwick.ac.uk

## Abstract

Network data are ubiquitous in our daily life, containing rich but often sensitive information. In this paper, we expand the current static analysis of privatised networks to a dynamic framework by considering a sequence of networks with potential change points. We investigate the fundamental limits in consistently localising change points under both node and edge privacy constraints, demonstrating interesting phase transition in terms of the signal-to-noise ratio condition, accompanied by polynomial-time algorithms. The private signal-to-noise ratio conditions quantify the costs of the privacy for change point localisation problems and exhibit a different scaling in the sparsity parameter compared to the non-private counterparts. Our algorithms are shown to be optimal under the edge LDP constraint up to log factors. Under node LDP constraint, a gap exists between our upper bound and lower bound and we leave it as an interesting open problem, echoing the challenges in high-dimensional statistical inference under LDP constraints.

## 1 Introduction

Numerous application areas and everyday life routinely generate network data, which contain valuable but often sensitive information [e.g. 33, 31]. Understanding the underlying patterns of network data while preserving individuals' privacy is crucial in modern data analysis. Several attempts have been made, but mostly focus on studying single snapshots of networks (a.k.a. static networks) and/or subject to central differential privacy constraint, where a central data curator is allowed to handle raw information from all individuals [e.g. 21, 22, 23, 9, 30]. In this paper, we are instead concerned with understanding the dynamics of a sequence of networks (a.k.a. dynamic networks), under local privacy constraints (LDP), where no one is allowed to handle the raw data of other individuals [e.g. 13, 17, 26, 40, 34].

Dynamic networks are usually in the form of a sequence of static networks, along a linear ordering, say time. In the dynamic networks studies, it is vital to capture the ever changing nature. A handy and useful way to model the changes is to assume that there exists a sequence of unknown time points, where the underlying distributions change abruptly [e.g. 37, 38]. These unknown time points are referred to as change points. Identifying change points helps to pinpoint important events, and more accurately estimate underlying distributions, which can be regarded as stationary between two consecutive change points. Dynamic networks change point analysis has demonstrated its success in climatology [e.g. 28], crime science[e.g. 6] and neuroscience [e.g. 10, 29], to name but a few.

36th Conference on Neural Information Processing Systems (NeurIPS 2022).

Despite the growing popularity in studying dynamic networks, we have witnessed a vacuum in estimating change points while preserving data owners' privacy. Having said this, a line of attack has been made to analyse static network data under LDP constraints, where only data owners have access to their individual raw data [e.g. 34, 40]. The private analysis of network data is complicated by the fact that different LDP conditions are required depending on the information that one wants to protect. For example, in a relationship network, users may want to protect their edge information, i.e. whether they are connected to someone else or not. As we argue in Section 2.1, formalisation to protect such information should require minimal trust between users due to the symmetric nature of network data. In a recommending system network, users may want to protect their entire connection portfolio, representing the purchase status of a user over a collection of products. In a brain imaging network, a patient may even prefer protecting the entire network from adversarial inference attacks.

In view of the aforementioned state of the art, we list the contributions of this paper below.

● To our best knowledge, this is the first time investigating change point localisation in dynamic networks, under LDP constraints. We consider two dynamic network models, where a sequence of sensitive networks are generated from (a) inhomogeneous Bernoulli networks (IBN) (Definition 1) and (b) bipartite networks with possibly dependent Bernoulli entries (Definition 2). Multiple change points of the raw network distributions are allowed. To tailor to the network models, we consider two forms of privacy requirements - edge LDP and node LDP (See Section 2).

● For dynamic IBNs under edge LDP, we show a phase transition in terms of the signal-to-noise ratio, partitioning the whole parameter space into two parts: (1) the infeasibility regime where no algorithm is expected to provide consistent change point estimators and (2) the regime where a computationally-efficient algorithm is shown to output consistent estimators. The importance of this phase transition is twofold: (1) The transition boundary is different from its counterpart in the non-private case [38], quantifying the cost of preserving edge privacy in localising change points. (2) We show that a simple randomised response [39] based privacy mechanism is minimax rate-optimal for the purpose of change point localisation.

● For bipartite networks under node LDP, we derive an infeasibility regime which is different from that under the edge LDP. This fundamental difference quantifies the difference between these two different LDP constraints, and can be used to help practitioners designing data collection mechanisms. We adopt a privacy mechanism proposed in [13], together with a change point estimation routine, providing a consistent change point estimator. Supported by a minimax lower bound result, our estimator is shown to be minimax rate optimal when the number of columns is of constant order. When the number of columns is allowed to diverge, a gap between our lower and upper bounds exists. This echos the well-identified challenges in the high-dimensional privacy research [e.g. 13, 14]. We contribute a high-dimensional network example along with in-depth discussions.

**Notation** For any matrix $A \in \mathbb{R}^{M \times N}$, let $A_{ij}$ be the $(i, j)$-th entry of $A$, $A_i \in \mathbb{R}^N$ be the $i$-th row of $A$, $A^\top$ be the transpose of $A$, $\|A\|_\infty = \max_{1 \leq i \leq M, 1 \leq j \leq N} |A_{ij}|$ and $\|A\|$ denote the operator norm of $A$. For any matrix $B \in \mathbb{R}^{M \times N}$, let $(A, B) = \sum_{1 \leq i \leq M, 1 \leq j \leq N} A_{ij} B_{ij}$ and $\|A\|_\mathrm{F} = \sqrt{(A, A)}$ be the Frobenius norm of $A$. For any vector $v \in \mathbb{R}^p$, let $\|v\|_1, \|v\|_2, \|v\|_\infty$ be the $\ell_1$, $\ell_2$ and $\ell_\infty$ vector norms respectively. For any set $S$, let $|S|$ be the cardinality of $S$. Let $S = S' \sqcup S''$, if $S' \cup S'' = S$ and $S' \cap S'' = \emptyset$. Let $1\{\cdot\}$ be the indicator function only taking values in $\{0, 1\}$. For two any functions of $T$, say $f(T)$ and $g(T)$, we write $f(T) \gtrsim g(T)$ if there exists constants $C > 0$ and $T_0$ such that $f(T) \geq Cg(T)$ for any $T \geq T_0$, and write $f(T) \asymp g(T)$ if $f(T) \gtrsim g(T)$ and $f(T) \lesssim g(T)$.

## 1.1 Problem setup

We consider two parallel models of dynamic networks. The first one is built upon an IBN model, which covers a wide range of models for undirected networks, including the Erdős–Rényi random graph [16], the stochastic block model [18] and the random dot product graph [e.g. 3], among others.

**Definition 1** (Inhomogeneous Bernoulli network, IBN). *A network with node set $\{1, \ldots, n\}$ is an inhomogeneous Bernoulli network if its adjacency matrix $A \in \mathbb{R}^{n \times n}$ satisfies that $A_{ij} = A_{ji} = 1\{nodes\ i, j\ are\ connected\ by\ an\ edge\}$ and $\{A_{ij}, i \leq j\}$ are independent Bernoulli random variables with $\mathbb{E}(A_{ij}) = \Theta_{ij}$.*

The second model considered is a bipartite IBN with possibly correlated entries within each row of the biadjacency matrix. See [2] for more discussions on bipartite networks.

**Definition 2** (Bipartite IBN). *A network with node set $\{1, \ldots, n_1 + n_2\} = V_1 \sqcup V_2$, $|V_1| = n_1$ and $|V_2| = n_2$, is a bipartite IBN if its biadjacency matrix $A \in \mathbb{R}^{n_1 \times n_2}$ satisfies the following. (1) For $i \in V_1$ and $j \in V_2$, $A_{ij} = 1\{$nodes $i, j$ are connected $\}$. (2) For any $i_1, i_2 \in V_1$, $i_1 \neq i_2$, $\{A_{i_1,j}, j = 1, \ldots, n_2\}$ and $\{A_{i_2,j}, j = 1, \ldots, n_2\}$ are independent. (3) For any $i \in V_1$, $j \in V_2$, $A_{ij}$ is a Bernoulli random variable with $\mathbb{E}(A_{ij}) = \Theta_{ij}$.*

Bipartite IBNs are often used in the recommending system, where each $i \in V_1$ represents a user and each $j \in V_2$ represents a product [e.g. 27, 20]. An important difference between Definitions 1 and 2 is that, in Definition 1 all entries are assumed to be independent, while in Definition 2, entries within the same row are allowed to be arbitrarily dependent. Dependence in networks are common in practice, for example the control-flow graph considered in [42] where $V_1$ corresponds to the set of users and each node in $V_2$ corresponds to a component within some software application and the dependencies therein are due to the causality between nodes.

The change points are defined formally in Assumption 1 where the magnitude of the distributional change is measured by the normalised Frobenius norm. The choice of Frobenius norm captures both dense and sparse changes in the network structure, see [38].

**Assumption 1.** *Let $\{A(t)\}_{t=1}^T \subset \{0,1\}^{n_1 \times n_2}$ be an independent sequence of adjacency matrices of IBNs defined in Definition 1 (in which case $n_1 = n_2 = n$) or biadjacency matrices of bipartite IBNs defined in Definition 2, with $\mathbb{E}\{A(t)\} = \Theta(t)$. Assume that there exist $\{\eta_1, \ldots, \eta_K\} \subset \{2, \ldots, T\}$, with $1 = \eta_0 < \eta_1 < \ldots < \eta_K \leq T < \eta_{K+1} = T + 1$, such that $\Theta(t) \neq \Theta(t - 1)$, if and only if $t \in \{\eta_1, \ldots, \eta_K\}$.*

*Let $\Delta = \min_{k=1}^{K+1}(\eta_k - \eta_{k-1})$ be the minimal spacing and $\kappa_0 = \min_{k=1}^K \|\Theta(\eta_k) - \Theta(\eta_k - 1)\|_{\mathrm{F}}/(\sqrt{n_1 n_2}\rho)$ be the minimal jump size, where $\rho = \max_{t=1}^T \|\Theta(t)\|_\infty$ denotes the entry-wise sparsity.*

For both models, under privacy constraints to be discussed in Section 2, our goal is to construct consistent estimators $\{\widehat{\eta}_k\}_{k=1}^{\widehat{K}}$ of $\{\eta_k\}_{k=1}^K$. To be specific, $\{\widehat{\eta}_k\}_{k=1}^{\widehat{K}}$ is said to be consistent if $\Delta^{-1} \max_{k=1}^K |\widehat{\eta}_k - \eta_k| \to 0$ and $\widehat{K} = K$ holds with probability tending to 1, as the sample size $T$ grows unbounded.

Lastly, we note that in statistical network analysis, when allowing for entry-wise sparsity, it is usually assumed that $\rho \geq \log(n)/n$ [e.g. 38] to ensure there are sufficiently many observed edges. However, We do not impose lower bounds on $\rho$ in Assumption 1, since to preserve privacy, the expectations of privatised network entries are inflated by a factor of the privacy level $\alpha \in (0, 1)$. Let $\rho'$ be the sparsity parameter of the privatised networks. Such inflation automatically ensures that $\rho' \geq \log(n)/n$, for any $\rho \in [0, 1]$ and $n > 1$ (See the proof of Theorem 3).

## 2 Network local differential privacy

To formalise different network LDP notions, we first recall a general definition of LDP. A private mechanism is a conditional distribution, which conditional on raw data, outputs privatised data. For a pre-specified privacy level $\alpha \geq 0$, a random object $Z_i$ taking values in $\mathcal{Z}$ is a non-interactive[1] $\alpha$-LDP version of raw data $X_i$, if for any raw data $x$ and $x'$, any measurable set $S \subset \mathcal{Z}$, it holds that

$$Q_i(Z_i \in S | X_i = x)/Q_i(Z_i \in S | X_i = x') \leq e^\alpha. \tag{1}$$

A privacy mechanism is $\alpha$-LDP if all output $Z_i$'s are $\alpha$-LDP. We focus on the regime $\alpha \in (0, 1)$, where the effect of privacy is the strongest and is often the regime of primary interest [e.g. 12, 13, 5, 36].

In view of (1), the LDP constraint ensures that each individual $i$ only has access to their own raw data. To be specific, the privatised $Z_i$ only depends on the raw data $X_i$. As for network data, to impose LDP, it is crucial to formalise what a unit of information includes and who are the owners of each unit of information. In the rest of this section, we consider two cases arising from different application backgrounds.

---

[1]In this paper, we only consider non-interactive privacy mechanisms throughout and will call it privacy mechanism when there is no concern of ambiguity. For more general interactive privacy mechanisms, see [13].

## 2.1 Edge local differential privacy in inhomogeneous Bernoulli networks

In epidemiological studies on sexually transmitted diseases, network data are formed by edges linking sexual partners [e.g. 35]. A natural choice of information unit is the existence of sexual relationship among subjects. Due to the sensitivity of such data, one may wish to consider all parties involved to be the owners of a potential link. Inspired by such applications, we formalise the edge LDP in Definition 3.

**Definition 3** (Edge $\alpha$-LDP). *We say that the privacy mechanism $Q = \{Q_{ij}^{(t)}, 1 \leq i \leq j \leq n, 1 \leq t \leq T\}$ is edge $\alpha$-LDP, if for any integer $1 \leq t \leq T$, any integer pair $1 \leq i \leq j \leq n$, any measurable set $S \subset \mathcal{Z}$ and any $x, x' \in \{0, 1\}$, it holds that*

$$Q_{ij}^{(t)}(Z_{ij}(t) \in S | A_{ij}(t) = x) / Q_{ij}^{(t)}(Z_{ij}(t) \in S | A_{ij}(t) = x') \leq e^{\alpha}. \tag{2}$$

Definition 3 is seemingly stricter than some existing edge LDP notions [e.g. 34, 40], where, instead of (2), it is required that for any $i \in \{1, \ldots, n\}$ and any $x, x' \in \mathbb{R}^n$ with $\|x - x'\|_1 = 1$,

$$Q_i^{(t)}(Z_i(t) \in S | A_i(t) = x) / Q_i^{(t)}(Z_i(t) \in S | A_i(t) = x') \leq e^{\alpha}. \tag{3}$$

It is clear that applying any mechanisms that satisfy (2) to each entry of $A_i(t)$ guarantees (3). While appropriate for some settings, the edge LDP defined in (3) possesses some caveats [19], listed below.

• The same piece of information is recorded in the adjacency matrix twice, i.e. $A_{ij}(t) = A_{ji}(t)$, and these two records are treated as being potentially different. The same quantity is thus randomised twice, leading to some inefficiency.

• As a more damning issue, mechanisms satisfying (3) may require trust between nodes. If a node does not follow the protocol correctly, or their data are intercepted, they may reveal information on other nodes in the network. This is not the case with LDP mechanisms in other settings, where the privacy of an individual is guaranteed regardless of the behaviour of other individuals.

Our definition (2) does not suffer either of these drawbacks since we only privatise the upper triangular part of the adjacency matrix, and to privatise each edge between two nodes, (2) implicitly requires that both parties to agree on their status and the privatised result so that the trust issue can be prevented.

## 2.2 Node local differential privacy in bipartite inhomogeneous Bernoulli networks

In a Netflix data set, one may model the viewing history by a dynamic bipartite IBN, where each row represents a user, each column represents a movie and each snapshot of network gathers the viewing information within a short time frame. It is reasonable to consider an information unit to be the viewing history of a user within a time frame, which is a row in a biadjacency matrix. Inspired by such applications, we formalise the bipartite node LDP in Definition 4.

**Definition 4** (Bipartite node $\alpha$-LDP). *We say that the privacy mechanism $Q = \{Q_i^{(t)}, 1 \leq i \leq n_1, 1 \leq t \leq T\}$ is bipartite node $\alpha$-LDP, if for any integer $1 \leq t \leq T$, any integer $1 \leq i \leq n_1$, any measurable set $S \subset \mathcal{Z}$ and any $x, x' \in \{0, 1\}^{n_2}$, it holds that*

$$Q_i^{(t)}(Z_i(t) \in S | A_i(t) = x) / Q_i^{(t)}(Z_i(t) \in S | A_i(t) = x') \leq e^{\alpha}. \tag{4}$$

Different notions of node LDP have been studied in the literature. Our definition (4) is consistent with [e.g. 34, 40] while some adopt the definition inherited from central DP allowing the neighbouring networks to have different dimension by either inclusion and deletion of one node [23, 11]. Several works consider the same constraint as (4) under the name user-level LDP [e.g. 24, 43] for different learning tasks.

One appealing feature of bipartite graphs when considering node LDP is that the neighbouring data sets $x, x'$ can be protected independently for each node in $V_1$, whereas in a general graph, node LDP should account for the intrinsic symmetry of the adjacency matrix when defining neighbouring data sets [19]. Comparing the two LDP definitions we considered in this section, we see that in Definition 3 level $\alpha$ privacy is imposed to protect one edge, and in Definition 4 level $\alpha$ privacy is imposed to protect $n_2$ edges. For the same privacy parameter $\alpha$, node privacy is a much more stringent constraint than edge privacy [e.g. 34, 40, 19].

# 3 Fundamental limits in consistent change point localisation

Recall that our task is to understand how the underlying distributions of dynamic networks change, especially to provide consistent change point estimators defined in Section 1.1, under certain form of LDP constraints. Without the concern of privacy, dynamic IBN change point localisation is investigated in [38], where a scaling (namely the signal-to-noise ratio) is proposed to partition the whole parameter space into two regimes: a low signal-to-noise ratio regime (infeasibility regime) where no consistent estimator is guaranteed in a minimax sense, and a high signal-to-noise ratio regime where computationally-efficient algorithms are shown to produce consistent estimators. Recall the model parameters $\kappa_0$ the minimal jump size, $\rho$ the entry-wise sparsity of networks, $n$ the network size and $\Delta$ the minimal spacing. Without the presence of privacy constraints, the infeasibility regime [38] is

$$\kappa_0^2 \rho n \Delta \lesssim 1, \tag{5}$$

which will serve as the benchmark for us to quantify the cost of privacy.

**The first model** we study is a dynamic IBN model (Definition 1 and Assumption 1), which is identical to the one studied in [38]. Lemma 1 demonstrates an infeasiblity regime of localising change points in such a model under the edge $\alpha$-LDP defined in Definition 3.

**Lemma 1** (Edge $\alpha$-LDP). *Let $\{A(t)\}_{t=1}^{T} \subset \{0,1\}^{n \times n}$ be a sequence of adjacency matrices satisfying Assumption 1 with $K = 1$ and let $P_{\kappa_0,\Delta,n,\rho}^{T}$ denote their joint distribution. Consider the class of distributions*

$$\mathcal{P} = \{P_{\kappa,\Delta,n,\rho}^{T} : \kappa_0^2 \leq \min\{[68n\rho^2\Delta(e^\alpha - 1)^2]^{-1}, 1/4\}, \Delta \leq T/3\}.$$

*Let $\mathcal{Q}_\alpha^{\mathrm{edge}}$ denote the set of all non-interactive privacy mechanisms that satisfy the edge $\alpha$-LDP constraint in Definition 3, for $\alpha \in (0, \min\{1, (2\rho)^{-1}\})$. We have that*

$$\inf_{Q \in \mathcal{Q}_\alpha^{\mathrm{edge}}} \inf_{\hat\eta} \sup_{P \in \mathcal{P}} \mathbb{E}_{P,Q}|\hat\eta - \eta(P)| \geq \Delta/12,$$

*where $\eta(P)$ denotes the change point location specified by distribution $P$, the first infimum is taken over all possible non-interactive privacy mechanisms, the second infimum is taken over all measurable functions of the privatised data and the supremum is taken over all raw data's distributions in the class $\mathcal{P}$.*

Lemma 1 studies an LDP minimax lower bound in the framework put forward by [13]. It shows that for dynamic IBNs under edge $\alpha$-LDP, provided $\kappa_0^2\rho^2 n\Delta(e^\alpha - 1)^2 \asymp \kappa_0^2\rho^2 n\Delta\alpha^2 \lesssim 1$, the localisation error $\Delta^{-1}|\hat\eta - \eta(P)| \geq 1/12$. This leads to the infeasiblity regime

$$\kappa_0^2\rho^2 n\Delta\alpha^2 \lesssim 1. \tag{6}$$

Comparing (5) and (6), any distribution in the regime (5) also falls in the regime (6), implying that imposing edge $\alpha$-LDP enlarges the infeasibility regime and makes the localisation task harder. To be specific, the cost of preserving edge LDP comes from two fronts.

• The effective sample size is decreased from $\Delta$ to $\Delta\alpha^2$. LDP's impact on the effective sample size is commonly observed in the literature over a wide range of problems [e.g. 13, 7, 5, 25].

• A more interesting and problem-specific cost of LDP is reflected by the role of the sparsity parameter $\rho$, which power is raised to $\rho^2$ in (6) from $\rho$ in (5). Despite that networks have been studied under LDP constraints, such result is the first time seen. Similar effects have been observed in different problems under LDP constraint, including the impacts on dimensionality [e.g. 5] and smoothness levels [e.g. 25]. It is interesting to see that in a high-dimensional sparse network problem, this problem-specific cost of LDP appears on the sparsity parameter.

**The second model** we consider is a dynamic bipartite IBN model (Definition 2 and Assumption 1), the change point analysis of which is not seen in the literature, even without privacy concerns. In addition to the rows and columns of bipartite IBNs denoting different entities, which is different from well-studied network models, we also allow potentially arbitrary within-row dependence. Lemma 2 establishes an infeasiblity regime of localising change points in such a model under the bipartite node $\alpha$-LDP defined in Definition 4.

**Lemma 2** (Bipartite node $\alpha$-LDP). *Let $\{A(t)\}_{t=1}^T \subset \{0,1\}^{n_1 \times n_2}$ be a sequence of biadjacency matrices satisfying Assumption 1 with $K = 1$ and let $P_{\kappa_0,\Delta,n_1,n_2,\rho}^T$ denote their joint distribution. Consider the class of distributions*

$$\mathcal{P} = \{P_{\kappa_0,\Delta,n_1,n_2,\rho}^T : \kappa_0^2 \leq \min\{[20n_1^{1/2}\rho^2\Delta(e^\alpha - 1)^2]^{-1}, 1/4\}, \Delta \leq T/3\}.$$

*Let $\mathcal{Q}_\alpha^{\mathrm{node}}$ denote the set of all non-interactive privacy mechanisms that satisfy the bipartite node $\alpha$-LDP constraint in Definition 4, for $\alpha \in (0, \min\{1, (4\rho)^{-1}\})$. We have that*

$$\inf_{Q \in \mathcal{Q}_\alpha^{\mathrm{node}}} \inf_{\hat{\eta}} \sup_{P \in \mathcal{P}} \mathbb{E}_{P,Q}|\hat{\eta} - \eta(P)| \geq \Delta/12,$$

*where $\eta(P)$ denotes the change point location specified by distribution $P$.*

In an LDP minimax framework, Lemma 2 shows that provided $\kappa_0^2\rho^2 n_1^{1/2}\Delta\alpha^2 \lesssim 1$, the localisation error $\Delta^{-1}|\hat{\eta} - \eta(P)| \geq 1/12$. This leads to the infeasibility regime

$$\kappa_0^2\rho^2 n_1^{1/2}\Delta\alpha^2 \lesssim 1. \tag{7}$$

To compare (6) and (7), we first let $n_1 = n_2 = n$ in Lemma 2 for convenience. The infeasibility regime under the node LDP reads as $\kappa_0^2\rho^2 n^{1/2}\Delta\alpha^2 \lesssim 1$, which compared to (6) implies that the cost of node LDP is higher than the edge LDP. To further understand the difference between node LDP and edge LDP, we let $n = \sqrt{n_1 n_2}$ in Lemma 1. The infeasibility regime under the edge LDP reads as $\kappa_0^2\rho^2(n_1 n_2)^{1/2}\Delta\alpha^2 \lesssim 1$, which compared to (7) highlights the difference of $n_2^{1/2}$, an extra cost of dimensionality. The extra cost captures the difference between privatising vectors with possibly correlated entries under node LDP and privatising discrete values under edge LDP.

Lastly, we note that both proofs of Lemma 1 and Lemma 2 follow the convex Le Cam method [e.g. 41] where the key step is bounding the chi square distance between a privatised product distribution and mixture of privatised product distributions. For the node LDP case, the hard distributions exploit the dependence within rows of the biadjancency matrices.

## 4 Consistent private network change point algorithms

We have established infeasibility regimes of change point localisation tasks under different network LDP constraints in Section 3 and have understood how the privacy preservation makes the tasks fundamentally harder. In this section, we provide polynomial-time private algorithms to obtain consistent change point estimators outside of the infeasibility regimes. A private algorithm has two key ingredients: (1) a privacy mechanism and (2) an algorithm with privatised data as inputs. For the two models we consider in this paper, we adopt the same change point localisation algorithm, while using different privacy mechanisms.

The change point localisation algorithm we consider is the network binary segmentation (NBS) algorithm proposed and studied in [38]. It is shown that NBS provides consistent change point estimators without privacy concerns, under minimax optimal conditions. For completeness, we include NBS in Algorithm 1 and introduce the CUSUM statistic below. For any form of data $\{X_i\}_{i=1}^T$ and any integer triplet $0 \leq s < t < e \leq T$, the CUSUM statistic is defined as

$$\widetilde{X}^{(s,e)}(t) = \sqrt{\frac{e-t}{(e-s)(t-s)}} \sum_{i=s+1}^t X_i - \sqrt{\frac{t-s}{(e-s)(e-t)}} \sum_{i=t+1}^e X_i.$$

As pointed out in [38], two sequences of independent networks are required as inputs of Algorithm 1 in order to estimate the Frobenius norm of an IBN. In practice, one can split the data to even and odd indices to obtain two sequences of networks.

### 4.1 Edge $\alpha$-LDP

To privatise a dynamic IBN (Definition 1) under the edge $\alpha$-LDP, we apply the randomised response mechanism [39] independently to every edge. The privacy guarantee follows by virtue of the the randomised response mechanism [15]. To be specific, given data $\{A(t)\}_{t=1}^T \subset \{0,1\}^{n \times n}$, let

---

**Algorithm 1** Network Binary Segmentation. NBS$((s,e),\{(\alpha_m,\beta_m)\}_{m=1}^M,\tau)$

---

**INPUT:** $\{U(t)\}_{t=1}^T, \{V(t)\}_{t=1}^T \subset \mathbb{R}^{n_1 \times n_2}, \{(\alpha_m,\beta_m)\}_{m=1}^M \subset [0,T], \tau_1 > 0$
  **for** $m = 1,\ldots,M$ **do**
    $[s'_m,e'_m] \leftarrow [s,e] \cap [\alpha_m,\beta_m], (s_m,e_m) \leftarrow [s'_m + 64^{-1}(e'_m - s'_m), e'_m - 64^{-1}(e'_m - s'_m)]$
    **if** $e_m - s_m \geq 1$ **then**
      $b_m \leftarrow \arg\max_{t=s_m+1,\ldots,e_m-1}(\tilde{U}^{(s_m,e_m)}(t), \tilde{V}^{(s_m,e_m)}(t))$
      $a_m \leftarrow (\tilde{U}^{(s_m,e_m)}(b_m), \tilde{V}^{(s_m,e_m)}(b_m))$
    **else**
      $a_m \leftarrow -1$
    **end if**
  **end for**
  $m^* \leftarrow \arg\max_{m=1,\ldots,M} a_m$
  **if** $a_{m^*} > \tau$ **then**
    add $b_{m^*}$ to the set of estimated change points
    NBS$((s,b_{m*}),\{(\alpha_m,\beta_m)\}_{m=1}^M,\tau)$
    NBS$((b_{m*}+1,e),\{(\alpha_m,\beta_m)\}_{m=1}^M,\tau)$
  **end if**
**OUTPUT:** The set of estimated change points.

---

$\{U_{t,i,j}, 1 \leq i \leq j \leq n\}_{t=1}^T$ be independent Unif$[0,1]$ random variables that are independent of $\{A(t)\}_{t=1}^T$. For any $t \in \{1,\ldots,T\}$ and any integer pair $1 \leq i \leq j \leq n$, let the privatised data be $\{A'(t)\}_{t=1}^T \subset \{0,1\}^{n \times n}$ with

$$A'_{ij}(t) = A'_{ji}(t) = \begin{cases} A_{ij}(t), & U_{t,i,j} \leq e^\alpha/(1+e^\alpha), \\ 1 - A_{ij}(t), & \text{otherwise.} \end{cases} \tag{8}$$

Note that due to the symmetry of the networks, each edge is only privatised once. Despite that we are dealing with a high-dimensional, sparse dynamic IBN model, with potentially multiple change points, Theorem 3 below shows that this, arguably simplest privacy mechanism not only provides consistent change point estimators, but also is optimal in terms of the signal-to-noise ratio condition required.

**Theorem 3.** *Let $\{A(t)\}_{t=1}^T$ and $\{B(t)\}_{t=1}^T$ be two independent sequences of adjacency matrices satisfying Assumption 1. For an arbitrarily small $\xi > 0$ and an absolute constant $c_0 > 0$, assume that*

$$\kappa_0^2 \rho^2 n \Delta \alpha^2 \geq c_0 \log^{2+\xi}(T). \tag{9}$$

*Let $\{\widehat{\eta}_k\}_{k=1}^{\widehat{K}}$ be the output of the NBS algorithm, with inputs:*

- *$\{A'(t)\}_{t=1}^T$ and $\{B'(t)\}_{t=1}^T$, privatised version of $\{A(t)\}_{t=1}^T$ and $\{B(t)\}_{t=1}^T$ obtained through (8);*
- *$\{(\alpha_m,\beta_m)\}_{m=1}^M$, random intervals whose end points are drawn independently and uniformly from $\{1,\ldots,T\}$ such that $\max_{m=1}^M(\beta_m - \alpha_m) \leq C_R\Delta$, for some constant $C_R > 3/2$; and* •
*tuning parameter $\tau$ satisfying $c_1 n \log^{3/2}(T) < \tau < c_2 \kappa_0^2 n^2 \rho^2 \Delta \alpha^2$, where $c_1, c_2 > 0$ are absolute constants.*

*It holds with probability at least $1 - \exp\{\log(T/\Delta) - c_3 M\Delta/T\} - c_4 T^{-c_5}$ that*

$$\widehat{K} = K \quad and \quad \max_{k=1}^K |\widehat{\eta}_k - \eta_k| \leq c_6 \log(T)\{\sqrt{\Delta}/(\kappa_0 n\rho\alpha) + \sqrt{\log(T)}/(\kappa_0^2 \rho^2 n\alpha^2)\},$$

*where $c_3, c_4, c_5, c_6 > 0$ are absolute constants.*

Theorem 3 shows that, provided $M \gtrsim T\Delta^{-1}\log(T/\Delta)$, it holds with probability tending to one,

$$\Delta^{-1} \max_{k=1}^K |\widehat{\eta}_k - \eta_k| \lesssim \Delta^{-1}\log(T)\{\sqrt{\Delta}/(\kappa_0 n\rho\alpha) + \sqrt{\log(T)}/(\kappa_0^2 \rho^2 n\alpha^2)\} \to 0, \tag{10}$$

where the second inequality is due to (9). Recalling the consistency definition in Section 1.1, (10) implies the consistency of NBS with randomised response privacy mechanism under edge $\alpha$-LDP.

In view of the condition (9) and the edge LDP infeasibility regime (6), up to a logarithmic factor, we unveil a phase transition with boundary $\kappa_0^2 \rho^2 n \Delta \alpha^2 \asymp 1$ and show that the randomised response mechanism is optimal in the minimax sense. This is conceptually interesting since, as pointed out in [34], the privatised network obtained by (8) leads to a dense graph even though the original graph may be sparse and therefore does not represent the original graph well. However, our result shows that this is the best one can do for change point localisation, at least among non-interactive mechanisms.

## 4.2 Bipartite node $\alpha$-LDP

To privatise a dynamic bipartite IBN (Definition 2) under the bipartite node $\alpha$-LDP, we apply the privacy mechanism developed in Duchi et al. [12, 13] for privatising vectors with bounded $\ell_\infty$ norm to each row of the biadjacency matrices. This privacy mechanism has been used in the analysis of mean estimation [e.g. 13], nonparametric density estimation [e.g. 13, 25] and exact support recovery [e.g. 8] problems under LDP.

Given data $\{A(t)\}_{t=1}^T \subset \{0,1\}^{n_1 \times n_2}$, let $\{U_{t,i}\}_{t=1,i=1}^{T,n_1}$ be independent Unif$[0,1]$ random variables that are independent of $\{A(t)\}_{t=1}^T$ and let $\{\tilde{A}_{ij}(t)\}_{t=1,i=1,j=1}^{T,n_1,n_2}$ be random variables satisfying

$$\mathbb{P}\{\tilde{A}_{ij}(t) = 1 | A_{ij}(t)\} = 1 - \mathbb{P}\{\tilde{A}_{ij}(t) = -1 | A_{ij}(t)\} = \{1 + A_{ij}(t)\}/2.$$

Let

$$B = C_{n_2}(e^\alpha + 1)/(e^\alpha - 1) \quad \text{with} \quad C_{n_2}^{-1} = \begin{cases} \frac{1}{2^{n_2-1}}\binom{n_2-1}{(n_2-1)/2}, & n_2 \bmod 2 \equiv 1, \\ \frac{1}{2^{n_2-1}+\frac{1}{2}\binom{n_2}{n_2/2}}\binom{n_2-1}{n_2/2}, & n_2 \bmod 2 \equiv 0. \end{cases}$$

The privatised data $\{A'(t)\}_{t=1}^T \subset \{0,1\}^{n_1 \times n_2}$ are obtained by sampling

$$A_i'(t) \sim \begin{cases} \text{Unif}\left(z \in \{B, -B\}^{n_2} | \sum_{j=1}^{n_2} z_i \tilde{A}_{ij}(t) \geq 0\right), & U_{t,i} \leq e^\alpha/(1 + e^\alpha), \\ \text{Unif}\left(z \in \{B, -B\}^{n_2} | \sum_{j=1}^{n_2} z_i \tilde{A}_{ij}(t) \leq 0\right), & \text{otherwise.} \end{cases} \tag{11}$$

Note that $\|A_i(t)\|_\infty = 1$ for any $i = 1, \ldots, n_1$ and $t = 1, \ldots, T$. Applying (26) in [13] with $d = n_2$ guarantees that $A_i'(t)$ is an $\alpha$-private version of $A_i(t)$ and therefore satisfies the bipartite node $\alpha$-LDP constraint. In Theorem 4, we demonstrate that NBS with inputs obtained through (11) is consistent in localising change points under bipartite node $\alpha$-LDP constraint.

**Theorem 4.** *Let $\{A(t)\}_{t=1}^T$ and $\{B(t)\}_{t=1}^T$ be two independent sequences of biadjacency matrices satisfying Assumption 1. For an arbitrarily small $\xi > 0$ and an absolute constant $c_0 > 0$, assume that*

$$\kappa_0^2 \rho^2 \min\{\sqrt{n_1/n_2}, n_1/n_2\}\Delta\alpha^2 \geq c_0 \log^{2+\xi}(Tn_1n_2). \tag{12}$$

*Let $\{\widehat{\eta}_k\}_{k=1}^{\widehat{K}}$ be the output of the NBS algorithm with inputs:*

*• $\{A'(t)\}_{t=1}^T$ and $\{B'(t)\}_{t=1}^T$, privatised version of $\{A(t)\}_{t=1}^T$ and $\{B(t)\}_{t=1}^T$ obtained through (11);*
*• $\{(\alpha_m, \beta_m)\}_{m=1}^M$, random intervals whose end points are drawn independently and uniformly from $\{1, \ldots, T\}$ such that $\max_{m=1}^M(\beta_m - \alpha_m) \leq C_R\Delta$, for some constant $C_R > 3/2$; and • tuning parameter $\tau$ satisfying $c_1 n_2\alpha^{-2}\log^2(Tn_1n_2)\max\{\sqrt{n_1n_2}, n_2\} < \tau < c_2\kappa_0^2 n_1n_2\rho^2\Delta$, where $c_1, c_2 > 0$ are absolute constants.*

*It holds with probability at least $1 - \exp\{\log(T/\Delta) - c_3 M\Delta/T\} - c_4 T^{-c_5}$ that $\widehat{K} = K$ and*

$$\max_{k=1}^K |\widehat{\eta}_k - \eta_k| \leq c_6 \log(Tn_1n_2)\left(\frac{\sqrt{\Delta}}{\kappa_0\rho\alpha}\sqrt{\frac{n_2}{n_1}} + \frac{\log(Tn_1n_2)}{\rho^2\alpha^2\kappa_0^2}\max\left\{\sqrt{\frac{n_2}{n_1}}, \frac{n_2}{n_1}\right\}\right),$$

*where $c_3, c_4, c_5, c_6 > 0$ are absolute constants.*

Theorem 4 shows that, provided $M \gtrsim T\Delta^{-1}\log(T/\Delta)$, NBS with privatised inputs through channel (11) is consistent. When $n_2 \asymp 1$, the signal-to-noise ratio condition (12) and the infeasibility regime (7) demonstrate a phase transition with boundary $\kappa_0^2\rho^2 n_1^{1/2}\Delta\alpha \asymp 1$, up to a logarithmic factor. When $n_2$ is allowed to diverge, a gap between the infeasibility regime (7) and (12) - the regime where our proposed method is deemed to be consistent - emerges. The larger $n_2$ is, the larger the gap is. It is interesting to understand further what happens within the gap and we leave this as an open problem, which echos the challenging problems in high-dimensional statistical inference under LDP.

Despite this open problem, we would like to point out that the condition (12) already exhibits some fundamental difference between change point localisation and mean estimation problems. To make this clear, we simplify the model by considering only one change point occurs at $\Delta$ and set $n_1 = n_2 = n$. Then, for each $i \in \{1, \ldots, n\}$, $\{A_i(t)\}_{t=1}^\Delta$ can be regarded as $\Delta$ i.i.d random vectors

satisfying that $A_{ij}(t) \sim \text{Ber}(\theta_{ij})$. Let $\theta_i = \{\theta_{ij}\}_{j=1}^n \in [0,1]^n$. The fundamental limit of estimating $\theta_i$ is studied in [14]. With $\Delta$ i.i.d samples, their Corollary 3 gives

$$\inf_{\substack{\text{privacy} \\ \text{mechanism } Q}} \inf_{\widehat{\theta}} \sup_{P_{\theta_i}} \mathbb{E}\|\widehat{\theta} - \theta_i\|_2^2 \gtrsim n^2/(\Delta\alpha^2), \tag{13}$$

when $\alpha \lesssim 1$. Now, write $\{\theta_i'\}_{i=1}^{n_1}$ for the row means of the distribution after the change point and we have $\|\mathbb{E}\{A(\Delta)\} - \mathbb{E}\{A(\Delta+1)\}\|_F^2 = \sum_{i=1}^n \|\theta_i - \theta_i'\|_2^2$. Aggregating (13) over $i = 1, \ldots, n$, it is natural to expect one needs $\sum_{i=1}^{n_1} \|\theta_i - \theta_i'\|_2^2 \gtrsim n^3/(\Delta\alpha^2)$ in order to detect the change point at $\Delta$. However, under the same setup, Theorem 4 shows that consistent change point localisation can be achieved under (12), read as $\sum_{i=1}^{n_1} \|\theta_i - \theta_i'\|_2^2 \gtrsim n^2/(\Delta\alpha^2)$, up to logarithmic factors.

This discrepancy between $n^3$ and $n^2$ suggests that the gap between our upper bound and lower bound cannot be closed by straightforwardly applying techniques designed for establishing lower bounds in mean estimation problems. It also illustrates the general wisdom in the change point literature that, one may afford to sacrifice some accuracy in estimating the underlying distributions, if the goal is just to estimate the change points [e.g. 32].

To conclude this section, we would like to present some result of independent interest. It is studied in the existing literature [Appendix I.3 in 13] that the privatised output from (11) is unbiased, i.e. $\mathbb{E}\{A_i'(t)\} = \mathbb{E}\{A_i(t)\}$, while the covariance structure of the privatised output is unknown. In Lemma 5, we carefully analyse the covariance matrix of the privatised output and provide an upper bound on its operator norm. Due to its independent interest, we denote the raw data vector as $V = (V_i) \in \mathbb{R}^d$ and denote its privatised output obtained through (11) as $Z = (Z_i) \in \mathbb{R}^d$.

**Lemma 5.** *For any random vector $V \in \mathbb{R}^d$ with $\|V\|_\infty \le 1$, we have that*

$$\text{Var}(Z_i) = B^2 - \{\mathbb{E}(V_i)\}^2, \quad i = 1, \ldots, d; \tag{14}$$

*and*

$$\text{Cov}(Z_i, Z_j) = \begin{cases} -\mathbb{E}(V_i)\mathbb{E}(V_j), & d \bmod 2 \equiv 1, \\ -\mathbb{E}(V_i)\mathbb{E}(V_j) - \frac{C_{d,\alpha}}{d^{1/2}\alpha^2}\mathbb{E}(V_iV_j), & d \bmod 2 \equiv 0, \end{cases} \quad \forall i \ne j, \tag{15}$$

*where where $C_{d,\alpha} \in [C_0, C_1]$ for some absolute constants $C_1 > C_0 > 0$. Letting $\Sigma_Z$ be the covariance matrix of $Z$, it holds that*

$$\|\Sigma_Z\| \le \begin{cases} B^2 + \|\mathbb{E}(V)\|_2^2, & d \bmod 2 \equiv 1 \\ B^2 + \|\mathbb{E}(V)\|_2^2 + \frac{c\sqrt{d}}{\alpha^2}\sqrt{\max_{i,j}\mathbb{E}(V_iV_j)} & d \bmod 2 \equiv 0, \end{cases} \tag{16}$$

*where $c > 0$ is an absolute constant.*

## 5   Conclusion

In this paper, we studied network change point localisation problems under two forms of LDP constraints. New signal-to-noise conditions (9) and (12) are derived and by comparing with the non private counterpart, we quantify the cost of privacy as discussed in Sections 3 and 4. A change in the scaling of sparsity parameter in the private signal to noise conditions reveals a new challenge of learning dynamic networks with possibly sparse and correlated entries. The results are summarised in the table below, where for clarity we ignored logarithmic factors and consider $n_1 = n_2 = n$ in the bipartite node LDP case.

| No privacy [38] | Edge LDP [(6)&(9)] | Node LDP lower bound (7) | Node LDP upper bound (12) |
|---|---|---|---|
| $\kappa_0^2 \rho \asymp \frac{1}{n\Delta}$ | $\kappa_0^2 \rho^2 \asymp \frac{1}{n\Delta\alpha^2}$ | $\kappa_0^2 \rho^2 \lesssim \frac{1}{\sqrt{n}\Delta\alpha^2}$ | $\kappa_0^2 \rho^2 \gtrsim \frac{1}{\Delta\alpha^2}$ |

As for limitations, we have only considered non-interactive privacy mechanisms so far, and it would be interesting to further consider interactive ones. This could take two possible routes by including interactive mechanisms in the lower bound considerations and/or designing interactive mechanism

that improves on the current upper bound. Both directions present some challenges. From the lower bound perspective, even in our analysis of non-interactive mechanisms, we identify a technical challenge in controlling the $\chi^2$-divergence between mixtures of private distributions. Although some techniques have been developed for discrete distributions [e.g. 4, 1], the counterpart for high-dimensional discrete distributions is still largely unexplored. As for the upper bound, different entries in our network model follow different distributions, which is in sharp contrast to the usual i.i.d. case where interactive methods may be helpful. We therefore expect that allowing interaction within networks cannot improve the signal to noise ratio condition, while interaction across time points requires novel methodology that can handle temporal dependence, account for the decay of privacy and is suitable for the task of change point localisation. We leave that as our future work.

## Acknowledgements and Disclosure of Funding

The authors would like to thank Harry Giles for helpful discussion and suggesting the idea behind the proof of Theorem 3. TBB acknowledges the support of an Engineering and Physical Sciences Reseach Council (EPSRC) New Investigator Award EP/W016117/1. YY acknowledges the support of an EPSRC Standard Grant EP/V013432/1.

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
