# OpenReview forum: "Network change point localisation under local differential privacy"
_NeurIPS.cc/2022/Conference — NeurIPS 2022 Accept_

### Official Review · Reviewer_jC5X · 2022-06-30

**Rating:** 5
**Confidence:** 2
**Soundness:** 3 good
**Presentation:** 3 good
**Contribution:** 3 good

**Summary:**

In this paper, the authors consider consistently localizing change points under both node and edge local differential privacy (LDP). The authors proposed an algorithm under edge LDP that is shown to be optimal. For node LDP, the authors show there is a gap exists between upper and lower bounds.

**Questions:**

Q1. Is it possible to provide some empirical evaluation since you have provided an algorithm?

**Limitations:**

The authors adequately addressed the limitations and potential negative societal impact of their work.

**Strengths And Weaknesses:**

Strengths:
S1. The work seems solid. I didn't check for all the technical details, but the formal definitions and results seem to be finished well.
S2. The authors provide theoretical results to the problems.

Weaknesses:
W1. I did not fully get the reason why we should focus on these problems. I can understand network data is important, but I do not see why we focus on these two network models. The paper could benefit from better motivating why these problems are important in practice, more concrete examples of the two network definitions, and how close these two definitions model real-world scenarios.

---

> ### Author Response · Authors · 2022-07-31
> **Responses**
>
> Thank you very much for your appreciation and constructive comments. We reply to all your comments and questions point-by-point in the following. We have submitted revised main text file and supplementary materials.
>
> **On the network models**
>
> Thanks for giving us the opportunity to further elaborate on our motivation.  In this paper, we consider two types of network models. Both of them are inhomogeneous which means that each edge can be connected with different probabilities.
>
> The inhomogeneous Bernoulli network (IBN) detailed in Definition 1 is a fairly general definition, which includes a wide range of important network models as its special case, including Erd\H{o}s--R\'enyi random graph models, stochastic block models, mixed membership block models, random dot product graph models and general random dot product graph models. Our results can serve as a benchmark for future work on these models with special structures under local differential privacy constraints. In our paper, we focus on the case that IBNs are undirected networks, but the analysis can also be extended to directed networks. We use this model to emphasise the edge LDP concept, where a pair of nodes/individuals jointly own a unit of information. In application, sensitive networks such as sexually transmitted disease networks motivated us to adopt this model, as described at the beginning of Section 2.1.
>
> Another commonly encountered type of network appears in recommender systems, where a network encodes the relationship between a set of users and a set of commodities. In particular, each edge here is owned by a single user rather than a pair of users.  This is widely seen in the context of Netflix and Amazon types data, which motivated our study as described at the beginning of Section 2.2. This type of network can be modelled by a bipartite IBN model (Definition 2) where dependence among each user's choice over products can also be accommodated, and we argue in the last paragraph of Section 2.2 that it is convenient to consider this type of network under node LDP constraint which is stronger than the notion of edge LDP.
>
> **Numerical results**
>
> Following your suggestions, we provide some representative simulation results here, with more details in the revised supplementary materials, including plots of the results.
> - Setting.  We generate a sequence of $T$ independent IBNs or bipartite IBNs when considering node LDP, with the network size $n_1 = n_2 = n = 50$ and entrywise sparsity level $\rho = 0.4$.  There is one and only one change point with a balanced spacing, i.e.~the change point $\eta = \Delta = T/2$, where $\Delta$ is the minimal spacing. The expectations of the adjacency matrix before and after change point are $\Theta_{\text{pre}} = 0.1 \times 1_{n \times n}$ and $\Theta_{\text{post}} = 0.4\times 1_{n \times n}$, respectively, where $1_{n \times n} \in \mathbb{R}^{n \times n}$ has all entries being one. The normalised jump size is therefore $\kappa_0 = \|\Theta_{\text{post}} - \Theta_{\text{pre}}\|_{\text{F}}/(n\rho) = 0.75$. We consider different minimal spacing $\Delta$ and privacy budget $\alpha$ in the simulations.
> - Method. We use a simplified version of NBS algorithm based on the binary segmentation procedure. For small number of change points, our theory still holds for this computationally less demanding algorithm. The thresholding tuning parameter, above which change points are declared, is fixed to be $n\log^{1.5}(T)/10$, $n\log^{1.5}(T)/30$ and $n^2\log^2(n^2T)/10$ in the no privacy, edge LDP and node LDP cases.
> - Metric. Let the estimated set of change points be  $\set{\widehat{\eta}_i\}_\{i=1\}^{\hat\{K\}}$. We use $\max_i |\widehat{\eta}_i - \eta|/\Delta \in [0,1]$ to evaluate the performances. If no change point is returned, we output one.
> - Result.  The result is collected in the table below, each cell of which is the **median** over 100 repetitions.  Without any privacy constraint, i.e. using raw data, the change can be easily detected with $\Delta$ as small as $7$.  Imposing privacy guarantees require a larger $\Delta$ to consistently localise the change points. We can see that for the same sample size, the performance deteriorates as $\alpha$ decreases under edge LDP.  The node LDP is a more stringent requirement, compared to the edge LDP.  From the last three columns, we can see that with the same sample size the change can be perfectly localised with no error in the no privacy case, and very well localised under edge LDP with $\alpha = 0.1$, but to obtain a reasonable estimator, the node information can only be protected at level $\alpha = 1$.
>
> | $\Delta$ | 7 | 15 | 23|700 |1100 | 1500 |
> | ---- | --- | --- | ---- | ----- | ---- | ---- |
> |No privacy|0.143|0.091|0.091 | 0.000 | 0.000 | 0.000 |
> |Edge LDP $\alpha = 0.5$ |0.429|0.429|0.273|0.000|0.000|0.000|
> |Edge LDP $\alpha = 0.1$ | 1.000 |1.000 |1.000 | 0.018 | 0.007 | 0.003 |
> |Node LDP $\alpha = 1$ | 1.000 |1.000|1.000|0.897|0.175|0.084|

---

### Official Review · Reviewer_vH7B · 2022-07-11

**Rating:** 6
**Confidence:** 3
**Soundness:** 4 excellent
**Presentation:** 3 good
**Contribution:** 3 good

**Summary:**

This paper studies the problem of change point detection in sequences of networks/graphs with respect to local differential privacy constraints (respectively edge and node LDP). Under assumptions on the random structure of the graphs (IBN for instance), the authors establish minimax rates for change point detection using Le Cam’s method. Then, the authors show that using randomized response to obtain edge privacy alongside the NBS algorithm obtains the minimax rate for change point detection. The authors show similar results for node LDP for bipartite IBNs.

**Questions:**

As aforementioned, it may be valuable to introduce an informal theorem for your main results in the introduction of the paper. Other questions and comments can be found above.

**Limitations:**

The authors discuss the limitations of their paper in the conclusion. In particular, they point out that their paper and arguments do not go through in the setting of interactive LDP. Investigating the problem in this setting seems important, especially for network data where one would expect the edge/node inclusion probabilities to be correlated.

**Strengths And Weaknesses:**

Strengths: To the best of my knowledge this problem has not been attempted before. Moreover, the need for edge/node LDP is clearly presented in the paper with practical examples. The problem is concisely contained, as both lower and upper bounds rates are presented for the change point estimation. The paper was clearly presented, although presenting the key results/rates as an informal theorem e.g. in the introduction may be useful. There is also thorough discussion in the conclusion of why the techniques do not generalize to the interactive setting.

Weaknesses: The problem discussed and approaches used in the paper are not particularly novel, as the paper is concerned with taking existing statistical procedures and studying them in through the lens of (local) privacy. Moreover, it would be interested to have some experimentation to demonstrate the empirical performance of these estimators when being run on privatized networks. Lastly, the random structure assumed on the families of graphs (IBN, bipartite IBN) seems somewhat restrictive, especially when considering practical network data (e.g. is it realistic to assume that networks connections are independently generated?). Is there a way to incorporate correlation between the network edge indicators into this result (outside of that included in the bipartite IBN)?

---

> ### Author Response · Authors · 2022-07-31
> **Responses**
>
> Thank you very much for your appreciation and constructive comments. We reply to all your comments and questions point-by-point in the following. We have submitted revised main text file and supplementary materials.
>
> **Novelty**
>
> You are indeed right that the methods we adopt in the paper are from existing literature. We in fact see this as a strength rather than lack of novelty.  We adopt existing methods to study a new and important problem, while deriving new insights for existing popular privacy mechanism along the way (e.g. Lemma 5). The message that simple methods such as randomised response can be minimax optimal is useful for the community.
>
> **Numerical results**
>
> Following your suggestions, we provide some representative simulation results in the responses, with more details in the revised supplementary materials, including plots of the results.
>
> - Setting.  We generate a sequence of $T$ independent IBNs or bipartite IBNs when considering node LDP, with the network size $n_1 = n_2 = n = 50$ and entrywise sparsity level $\rho = 0.4$.  There is one and only one change point with a balanced spacing, i.e.~the change point $\eta = \Delta = T/2$, where $\Delta$ is the minimal spacing. The expectations of the adjacency matrix before and after change point are $\Theta_{\text{pre}} = 0.1 \times 1_{n \times n}$ and $\Theta_{\text{post}} = 0.4\times 1_{n \times n}$, respectively, where $1_{n \times n} \in \mathbb{R}^{n \times n}$ has all entries being one. The normalised jump size is therefore $\kappa_0 = \|\Theta_{\text{post}} - \Theta_{\text{pre}}\|_{\text{F}}/(n\rho) = 0.75$. We consider different minimal spacing $\Delta$ and privacy budget $\alpha$ in the simulations.
> - Method.  We use a simplified version of NBS algorithm based on the binary segmentation procedure. For small number of change points, our theory still holds for this computationally less demanding algorithm. The thresholding tuning parameter, above which change points are declared, is fixed to be $n\log^{1.5}(T)/10$, $n\log^{1.5}(T)/30$ and $n^2\log^2(n^2T)/10$ in the no privacy, edge LDP and node LDP cases.
> - Metric. Let the estimated set of change points be  $\set{\widehat{\eta}_i\}_\{i=1\}^{\hat\{K\}}$. We use $\max_i |\widehat{\eta}_i - \eta|/\Delta \in [0,1]$ to evaluate the performances.  If no change point is returned, we output one.
> - Result.  The result is collected in the table below, each cell of which is the **median** over 100 repetitions.  Without any privacy constraint, i.e. using raw data, the change can be easily detected with $\Delta$ as small as $7$.  Imposing privacy guarantees require a larger $\Delta$ to consistently localise the change points. We can see that for the same sample size, the performance deteriorates as $\alpha$ decreases under edge LDP.  The node LDP is a more stringent requirement, compared to the edge LDP.  From the last three columns, we can see that with the same sample size the change can be perfectly localised with no error in the no privacy case, and very well localised under edge LDP with $\alpha = 0.1$, but to obtain a reasonable estimator, the node information can only be protected at level $\alpha = 1$.
>
> | $\Delta$ | 7 | 15 | 23|700 |1100 | 1500 |
> | ----------- | ----------- | ----------- | -------- | ----------- | ----------- | ----------- |
> | No privacy | 0.143 | 0.091 | 0.091 | 0.000 | 0.000 | 0.000 |
> |Edge LDP $\alpha = 0.5$ |0.429|0.429|0.273|0.000|0.000|0.000|
> | Edge LDP $\alpha = 0.1$ | 1.000 |1.000 |1.000 | 0.018 | 0.007 | 0.003 |
> | Node LDP $\alpha = 1$ | 1.000 |1.000|1.000|0.897| 0.175 | 0.084|
>
> **Presentation**
>
> Thank you for the suggestion of adding some informal theorems in the introduction.  After some thought, we added a table summarising the main results in the Conclusion section rather than in the Introduction section, due to the difficulty of introducing any form of results without formal definitions of different privacy concepts.  The notions of local differential privacy on network data are still ambiguous in the existing literature and our paper contributes to this discussion.
>
> **Edge dependence**
>
> This is indeed a very valuable suggestion.  In the network statistics literature, even without privacy concerns, incorporating the dependence among edges is an important yet open task.  Due to the lack of a natural distance among edges, it is often hard to directly adopt the dependence assumptions used in time series or spatial statistics.  One way to impose edge dependence is to assume a hierarchy structure in modelling, for instance, random dot product graphs, where each node is associated with a latent position.  Statistical analysis involved usually requires singular value decomposition of the whole adjacency matrix and corresponding results under local differential privacy need to be developed. We will consider such models in our future work.

---

### Official Review · Reviewer_3623 · 2022-07-15

**Rating:** 7
**Confidence:** 5
**Soundness:** 4 excellent
**Presentation:** 4 excellent
**Contribution:** 4 excellent

**Summary:**

This paper studies the problem of change detection (or change point localization as mentioned by the authors) in dynamically evolving graphs subject to differential privacy. Focusing on two generative models, namely, inhomogeneous Bernoulli networks (IBNs) and bipartite IBNs, conditions are provided under which change localization is infeasible. Subsequently, a simple randomized response mechanism is studied (applied together with CUSUM type statistic for change detection) and analyzed.

**Questions:**

See comments above

**Limitations:**

Yes, limitations were appropriately discussed.

**Strengths And Weaknesses:**

+ This is a technically strong and very well written paper, which I enjoyed reading. I very much appreciated the care taken by the authors in describing the similarities/differences in techniques which they borrowed/adapted from prior works. In addition, some of the discussions along with the technical results (such as impact of privacy constraints on the graph sparsity) were quite intriguing. Overall, I believe this could be a useful contribution to the broader ML community working on the area of privacy preserving graph based algorithms.

Some minor weaknesses and suggestions for the authors:

- While I understand that this is a theoretically oriented paper, and the technical results are solid, I would have still liked to see some numerical results to actually see the finite sample performance of the proposed algorithm (RR + CUSUM) on some dynamic networks in practice.

- I would like to point out a somewhat related work on "Differentially Private Community Detection For Stochastic Block Models", ICML 2022, which also derives some tradeoffs between privacy and the phase-transition boundary (for community recovery). In contrast to the conclusions of this submission (which show that RR is almost optimal for edge LDP), the above reference shows that RR may not be a good choice for private community detection with edge DP, as it significantly increases the average node degree (and other mechanisms, such as exponential, and stability based methods can fare better and provide a better privacy-recovery tradeoff). This is a complementary viewpoint, which authors may wish to discuss in the paper, i.e., what (if any) are the limitations of RR based perturbations.

---

> ### Author Response · Authors · 2022-07-31
> **Responses**
>
> Thank you very much for your appreciation and constructive comments. We reply to all your comments and questions point-by-point in the following. We have submitted revised main text file and supplementary materials.
>
> **Numerical results**
>
> Following your suggestions, we provide some representative simulation results here, with more details in the revised supplementary materials, including plots of the results.
> - Setting. We generate a sequence of $T$ independent IBNs or bipartite IBNs when considering node LDP, with the network size $n_1 = n_2 = n = 50$ and entrywise sparsity level $\rho = 0.4$. There is one and only one change point with a balanced spacing, i.e.~the change point $\eta = \Delta = T/2$, where $\Delta$ is the minimal spacing. The expectations of the adjacency matrix before and after change point are $\Theta_{\text{pre}} = 0.1 \times 1_{n \times n}$ and $\Theta_{\text{post}} = 0.4\times 1_{n \times n}$, respectively, where $1_{n \times n} \in \mathbb{R}^{n \times n}$ has all entries being one. The normalised jump size is therefore $\kappa_0 = \|\Theta_{\text{post}} - \Theta_{\text{pre}}\|_{\text{F}}/(n\rho) = 0.75$. We consider different minimal spacing $\Delta$ and privacy budget $\alpha$ in the simulations.
> - Method. We use a simplified version of NBS algorithm based on the binary segmentation procedure. For small number of change points, our theory still holds for this computationally less demanding algorithm. The thresholding tuning parameter, above which change points are declared, is fixed to be $n\log^{1.5}(T)/10$, $n\log^{1.5}(T)/30$ and $n^2\log^2(n^2T)/10$ in the no privacy, edge LDP and node LDP cases.
> - Metric. Let the estimated set of change points be $\set{\widehat{\eta}_i\}_\{i=1\}^{\hat\{K\}}$. We use $\max_i |\widehat{\eta}_i - \eta|/\Delta \in [0,1]$ to evaluate the performances. If no change point is returned, we output one.
> - Result. The result is collected in the table below, each cell of which is the **median** over 100 repetitions. Without any privacy constraint, i.e. using raw data, the change can be easily detected with $\Delta$ as small as $7$. Imposing privacy guarantees require a larger $\Delta$ to consistently localise the change points. We can see that for the same sample size, the performance deteriorates as $\alpha$ decreases under edge LDP. The node LDP is a more stringent requirement, compared to the edge LDP. From the last three columns, we can see that with the same sample size the change can be perfectly localised with no error in the no privacy case, and very well localised under edge LDP with $\alpha = 0.1$, but to obtain a reasonable estimator, the node information can only be protected at level $\alpha = 1$.
>
> | $\Delta$ | 7 | 15 | 23|700 |1100 | 1500 |
> | ---- | --- | --- | ---- | ----- | ---- | ---- |
> |No privacy|0.143|0.091|0.091 | 0.000 | 0.000 | 0.000 |
> |Edge LDP $\alpha = 0.5$ |0.429|0.429|0.273|0.000|0.000|0.000|
> |Edge LDP $\alpha = 0.1$ | 1.000 |1.000 |1.000 | 0.018 | 0.007 | 0.003 |
> |Node LDP $\alpha = 1$ | 1.000 |1.000|1.000|0.897|0.175|0.084|
>
> **On the randomised response mechanism**
>
> Thanks for pointing out this relevant literature which we have cited in our revision.
>
> > [1] Mohamed, M. S., Nguyen, D., Vullikanti, A., & Tandon, R. (2022, June). Differentially Private Community Detection for Stochastic Block Models. In International Conference on Machine Learning (pp. 15858-15894). PMLR.
>
> In the following, we first comment on the use of randomised response (RR) mechanism in our paper and then discuss the connection with [1].
>
> In our paper, the simple RR mechanism is shown to be optimal in the edge privacy case, but *sub-optimal* in the node privacy case. The sub-optimality of RR in the node privacy case motivates our study of the more involved sampling mechanism in change point analysis. To be specific, using RR with parameter $\alpha/n_2$ to each entry of the network can achieve $\alpha$-level node privacy. We consider the simple case that $n_1 = n_2 = n$ and ignore logarithmic factors.  Similar arguments as those in the proof of Theorem 3 show that, to consistently localise change points, an RR-based method would require the condition
> \begin{align*}
>     \kappa_0^2 \rho^2 \gtrsim \frac{n}{\Delta\alpha^2}.
> \end{align*}
> Our proposed method improves the previous condition by a factor of $n$ to
> \begin{align*}
> \kappa_0^2 \rho^2 \gtrsim \frac{1}{\Delta\alpha^2}.
> \end{align*}
>
> In [1], RR is shown to be sub-optimal under edge *central* differential privacy to conduct community detection.  In our paper, RR is shown to be optimal under edge *local* differential privacy to conduct change point analysis.  Under central differential privacy, a central data curator has access to all the raw data.  As for local differential privacy, the raw data can only be accessed by the owners of data.  The existence of central data curator enables algorithms to borrow information from other data points, which changes the fundamental limit of the problem.

---

### Official Review · Reviewer_u65C · 2022-07-20

**Rating:** 6
**Confidence:** 2
**Soundness:** 3 good
**Presentation:** 2 fair
**Contribution:** 3 good

**Summary:**

The paper looks the problem of finding change point of networks sequences for two networks under edge and node LDP. For each setting, it shows the feasible regime and an algorithm that guarantee LDP.

**Questions:**

I mainly have some questions regarding the setup.

- I'm a bit confused by Assumption 1 & 2. I think they define the change point and some properties of the change point for the two networks. But isn't there a unified definition for any network? Why would we need two? (And, why are they "assumption" not "definition"?)

- Line 118 says we want the estimated change points to be close to the actual change points as sample size T grows. Is that under the assumption that the number of change points K does not grow with T?

- I think $A$ is used for both the adjacency matrix and the set $A \subset Z$. Maybe you can considering change the notation.

- For the privacy definitions in Section 2
  - what is Q? If it is a mechanism (as is stated in Line 144), I feel like its should be a function that operate on the input so the expression should be something like Z = Q(X)? But the way it is used in the formula make it seems like some probability measure.
  - what is X? Is that the adjacency matrix (so the same as A in Section 1)? And what is $X_i$? Is that one row of the adjacency matrix?
  - Regarding the edge-LDP definition:
    - You mentioned that some prior work is looking at $X_i$ instead of $X_{ij}$ and it requires trust between nodes. I didn't follow that part. Does that refer to the case where user $i$ is supposed to privatize the edge (or lack of edge) between $i$ and $j$? But if so, wouldn't (3) also have the same problem?
    - I don't quite follow the definition. I think what you mean is that for user i, even if every {i, j} changes, we would still be able to hide the change. But if so, isn't that the same as node-LDP?

**Limitations:**

Yes.

**Strengths And Weaknesses:**

Strength: The paper looks at an important problem which hasn't been worked on before.

Weakness: The privacy definitions is a bit hard to follow.

---

> ### Author Response · Authors · 2022-07-31
> **Responses**
>
> Thank you very much for your appreciation and constructive comments.  We reply to all your comments and questions point-by-point in the following.  We have submitted revised main text file and supplementary materials.
>
> **Presentation of privacy definitions**
>
> In the revision, we have changed the notations in Section 2 to improve readability.
>
> **Q1: Assumptions 1 & 2**
>
> You are indeed right that these two assumptions contain notation definitions and have substantial overlapping.  In the revision, we have merged these two assumptions following your suggestion.  We however keep the notation definition within the Assumption to save space.
>
> **Q2: the number of change points**
>
> We seek consistent estimators satisfying that
> \begin{align*}
>     \Delta^{-1} \max_{k = 1}^K |\widehat{\eta}_k - \eta_k| \to 0 \quad \mbox{and} \quad \widehat{K} = K,
> \end{align*}
> as the sample size $T$ grows unbounded.  The number of change point $K$ is also allowed to be a function of $T$, which means when $T$ grows unbounded, $K$ is also allowed to diverge.  The only condition required regarding $K$ can be inferred from the signal-to-noise ratio condition.  We use the edge privacy as an example to explain.  In eq.~(9) in Theorem 3, we require that
> \begin{align*}
>     \kappa_0^2 \rho^2 n \Delta \alpha^2 \geq c_0 \log^{2+\xi}(T).
> \end{align*}
> This means that we require the minimal spacing between two consecutive change points $\Delta$ to satisfy
> \begin{align*}
>     \Delta \gtrsim \log^{2+\xi}(T)/(\kappa_0^2 \rho^2 n \alpha^2).
> \end{align*}
> Since $K \leq T/\Delta$, this implies that the number of change points needs to be bounded by
> \begin{align*}
>     K \lesssim T \kappa_0^2 \rho^2 n \alpha^2/\log^{2+\xi}(T).
> \end{align*}
>
> **Q3: notation $A$**
>
> Thanks for pointing this out.  In the revision, we have changed the notation set $A$ to set $S$, while leaving $A$ to denote adjacency or biadjacency matrices.
>
> **Q4.1: privacy mechanism $Q$**
>
> The privacy mechanism $Q$ is a *randomised* mechanism so it is indeed a probability measure. It is in fact a conditional distribution, which, conditioning on the raw data, outputs privatised data.  The local differential privacy definition imposes directly on the likelihood ratio, as stated in eq.~(1) that
> \begin{align*}
>     Q_i(Z_i \in S | X_i = x)/Q_i(Z_i \in S | X_i = x') \leq e^{\alpha},
> \end{align*}
> where $X_i$ is a piece of generic notation that refers to the data point provided by user $i$.
>
> **Q4.2: notation $X$ in Section 2**
>
> Thanks for pointing this out.  The $X$ and its related quantities indeed correspond to the adjacency matrix, and we did use $X_i$ to denote the $i$-th row of the adjacency matrix. In the rebuttal revision, we have changed $X$ and its related quantities to $A$ and its related quantities to improve readability. We still use $X_i$ in equation (1), since it denotes a general LDP definition.
>
> **Q4.3.1: previous work and eq. (3) in our paper**
>
> Thanks for the opportunity to elaborate on this subtle point.  Previous work allows user $i$ to privatise the information owned jointly by themselves and other users, and this requires trust on user $i$ not to leak information about others. The core of local privacy is to allow minimal trust among users. To overcome this issue that a single corrupted user could affect other users' privacy guarantee, in our edge LDP part, we privatise each edge individually, with the consent/decision from both involved individuals $i$ and $j$.
>
> **Q4.3.2: edge vs. node privacy**
>
> One way to explain the difference between edge LDP and node LDP is to investigate how the privacy budget is spent. The edge LDP definition (Definition 3) requires that
> \begin{align*}
> Q_{ij}^{(t)}(Z_{ij}{(t)} \in S | A_{ij}{(t)} = x)/ Q_{ij}^{(t)}(Z_{ij}{(t)} \in S | A_{ij}{(t)} = x')\leq e^{\alpha}.
> \end{align*}
> for any $x, x' \in \{0,1\}$, which says each single edge is protected at level $\alpha$. In the bipartite node LDP definition (Definition 4), it requires $n_2$ edges to be protected at level $\alpha$. In other words, in the node privacy case, the privacy budget needs to be split among $n_2$ edges. To be specific, one can, although inefficiently, privatise each edge at level $\alpha/n_2$ to satisfy the node LDP definition at level $\alpha$.  Since smaller $\alpha$ means stronger privacy guarantee, each edge is *more* private under node LDP than edge LDP.  A more intuitive explanation is as follows.  Under edge LDP, one wants to make any two graphs differ by one edge look similar after privatisation.  Under node LDP, one wants to make any two graphs differ by one node's connection look similar after privatisation.  The level of similarity is quantified by $\alpha$.

---

### Meta-Review · Area_Chair_8zDb · 2022-08-29

**Recommendation:** Accept
**Confidence:** Certain

**Metareview:**

This paper considers the important problem of change point detection in networks under local differential privacy. The paper provides bounds for the problem under both node and edge privacy constraints. While the bounds in all cases are not matching the results are timely and will be interesting for several researchers.

**Award:**

No

---

### Decision · Program_Chairs · 2022-09-14

Accept